# Competitive protein recruitment in artificial cells
Thijs W. van Veldhuisen, Madelief A. M. Verwiel, Sebastian Novosedlik ⓘ , Luc Brunsveld ⓘ ✉ & Jan C. M. van Hest ⓘ ✉

Living cells can modulate their response to environmental cues by changing their sensitivities for molecular signals. Artificial cells are promising model platforms to study intercellular communication, but populations with such differentiated behavior remain underexplored. Here, we show the affinity-regulated exchange of proteins in distinct populations of coacervate-based artificial cells via protein-protein interactions (PPI) of the hub protein 14-3-3. By loading different coacervates with different isoforms of 14-3-3, featuring varying PPI affinities, a client peptide is directed to the more strongly recruiting coacervates. By switching affinity of client proteins through phosphorylation, weaker binding partners can be outcompeted for their 14-3-3 binding, inducing their release from artificial cells. Combined, a communication system between coacervates is constructed, which leads to the transport of client proteins from strongly recruiting coacervates to weakly recruiting ones. The results demonstrate that affinity engineering and competitive binding can provide directed protein uptake and exchange between artificial cells.

Living cells interact with each other by physical and chemical cues, which is essential for the coordination of cellular behavior in multicellular organisms[1], or for collective prokaryotic behavior such as quorum sensing[2]. Affinity-based protein recognition is one of the mechanisms of chemical signaling between cells, which can be highly differentiated. By regulating affinity for a ligand, different cells show different responsiveness to the same chemical cues and individual cells can change their affinity over time. One example of competitive protein recognition is the activation of the epidermal growth factor receptor (EGFR) upon binding of one of its seven ligands[3,4]. These ligands vary in affinity, and binding of distinct ligands leads to divergent downstream signaling responses.

Natural chemical communication between cells has inspired researchers to engineer intercellular signaling systems from the top-down[5,6] and bottom-up[6–8]. These synthetic cells have been constructed to secrete and/or sense small molecules[9–17] or macromolecules such as nucleic acids and proteins[18–21]. For example, communication between populations of artificial cells has been shown by exchange of DNA via programmed strand displacement reactions[18,19,22]. Moreover, Niederholtmeyer et al.[20] showed artificial quorum sensing of proteins by the exchange of an RNA polymerase for in vitro transcription/translation. However, differentiated behavior based on affinity regulation has remained underexplored in synthetic cell research.

We have previously developed an artificial cell platform based on synthetic (membranized) coacervates. With this system, we were able to demonstrate the specific recruitment of client proteins to, and the exchange of protein cargo between artificial cells, based on DNA-mediated protein shuttling[21]. Exchange or recruitment of client proteins in synthetic cells can also be governed by protein-protein interactions (PPIs)[23–27]. For us, in particular interactions of client proteins with the hub protein 14-3-3, a native protein which is important for the regulation of many signaling pathways[28,29], yielded specific uptake based on affinity for 14-3-3[26]. 14-3-3 generally binds to serine/threonine phosphorylated client proteins, although there are also examples of nonphosphorylated binding motifs in client proteins, such as the 14-3-3-binding motif of the bacterial toxin Exoenzyme S[30,31]. This versatile and tunable binding landscape of 14-3-3 with its binding partners now offers us an interesting platform to introduce affinity-regulated protein exchange between artificial cells.

In this work, we demonstrate differentiated behavior regarding the specific uptake and exchange of client proteins based on their interaction with the coacervate-incorporated natural hub protein 14-3-3. By loading 14-3-3 isoforms with different client binding affinities, and/or competing clients of 14-3-3 in distinct populations of coacervates, we can direct specific client proteins to coacervate populations that provide the highest affinity or availability of 14-3-3 (Fig. 1). This enables us to emulate naturally occurring affinity-regulated communication processes in an artificial cell platform.

Laboratory of Chemical Biology, Department of Biomedical Engineering and Institute for Complex Molecular Systems, Eindhoven University of Technology, Eindhoven, The Netherlands. ✉e-mail: l.brunsveld@tue.nl; j.c.m.v.hest@tue.nl

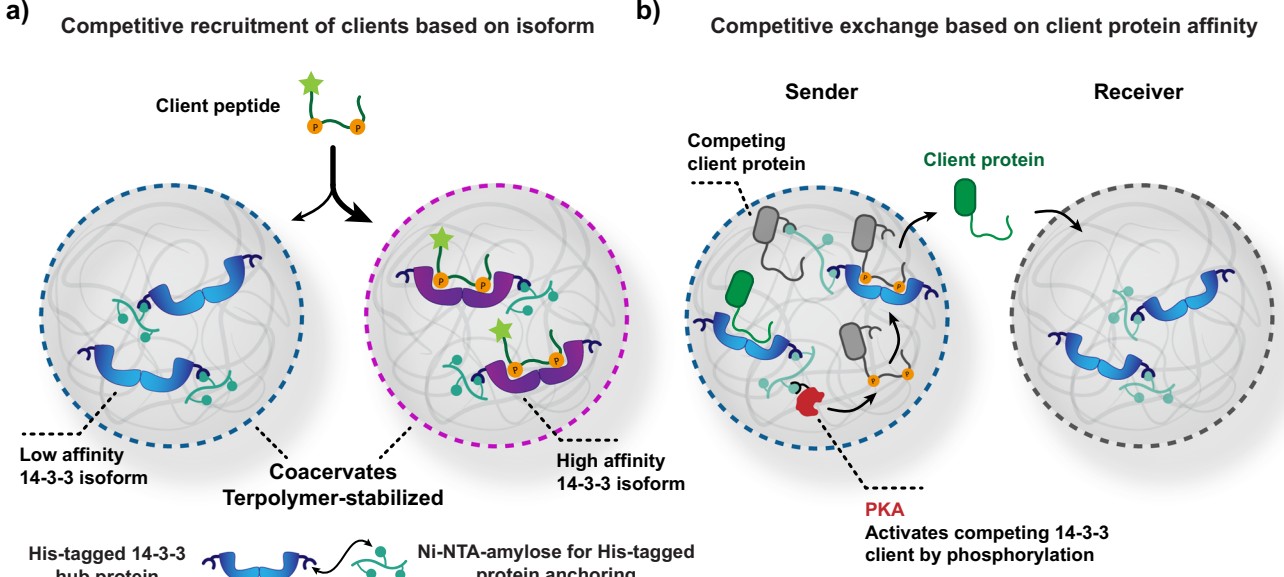

**Fig. 1 | Competitive protein-protein interactions drive the recruitment of clients to specific coacervate populations. a** Schematic overview of affinity-based uptake of a client peptide of the 14-3-3 hub protein in a specific coacervate population. Two isoforms of His-tagged 14-3-3 are anchored in separate terpolymer-stabilized coacervates by means of their interaction with Ni-NTA-amylose in the coacervates. **b** Schematic overview of inter-coacervate signaling based on 14-3-3 interactions. PKA phosphorylates a phosphorylation-dependent and coacervate-anchored client protein (competitor, gray), which displaces an initially bound, but mobile, client protein (moderate binder, green) and facilitates the release of the mobile client protein into bulk solution. Subsequently, the green client protein is taken up into the receiver population of coacervates. 14-3-3, PKA, and the phosphorylation-dependent client protein are immobilized in the coacervates by means of interactions between their His-tag and Ni-NTA-amyloses.

## Results

### Competitive client recruitment based on 14-3-3 isoform

14-3-3, a dimeric protein, has seven different human isoforms (β, γ, ε, ζ, η, σ, and τ), where σ is often the weakest binding isoform and γ is among the strongest binding isoforms with reported differences in affinity to client proteins of >50-fold[32–34]. We sought to demonstrate that when different isoforms of 14-3-3 were loaded in distinct coacervate populations, a client would be directed to the strongest recruiting population. The synthetic coacervates were formulated with positively charged quaternized amylose (Q-Am) and negatively charged carboxymethylated amylose (Cm-Am), with an overall excess of positive charge[35,36]. By incorporating nitrilo triacetic acid-modified amylose complexed with $Ni^{2+}$ (Ni-NTA-Am) in the coacervates, His-tagged 14-3-3 proteins could be embedded in a programmable manner mediated by their affinity for Ni-NTA-Am (Fig. 1). The coacervates were stabilized with a semipermeable triblock copolymer membrane[35], yielding stable droplets that do not fuse, enabling the study of the exchange of client proteins between coacervates[21].

Coacervates were prepared in a 2.5:0.8:0.2 charge ratio of Q-Am/Cm-Am/NTA-Am, which was found to give stable coacervates with efficient uptake of protein cargo. During preparation bulk concentrations of either 14-3-3σ (100 nM, Cy5-labeled) or 14-3-3γ (100 nM, DyLight 405-labeled) were added. The fluorescently labeled 14-3-3-His proteins were sequestered in the coacervates by interactions with the Ni-NTA-amylose, yielding a median local concentration of $35 \pm 11\,\mu M$ as determined by confocal microscopy (Supplementary Fig. 1). This local concentration represents a 350-fold enhancement of concentration compared to the bulk concentration of 100 nM and is comparable to the 14-3-3 loading that was determined in our previous work[26]. The fluorescein isothiocyanate (FITC)-labeled bivalent c-Raf pS233/pS259 (c-Raf pS) client peptide was added to the separate populations of coacervates at a concentration of 25 nM yielding a 14-3-3/c-Raf pS binding site ratio of 2:1 (Fig. 2a). The bulk affinity of the peptide for the two isoforms differs 40-fold, with a $K_D$ of $987 \pm 125$ nM for 14-3-3σ, and a $K_D$ of <25 nM for 14-3-3γ, as determined by fluorescence anisotropy (FA) assay (Supplementary Fig. 2). The local concentrations of both 14-3-3 isoforms in coacervates exceeded the $K_D$ of their interaction with c-Raf pS, and 14-3-3 was present in overall excess relative to c-Raf pS. Upon partitioning of the peptide into the coacervates it was therefore efficiently bound to 14-3-3, and it was hypothesized that this would lead to similar recruitment of c-Raf pS in single populations of coacervates containing either the strong or weak isoform.

Although we have previously used a bioluminescence assay for PPIs in coacervates[26], confocal microscopy is more suited for determining the localization of a client in a multi-population coacervate sample over time since it additionally provides spatial information and is not enzyme substrate-dependent. Confocal micrographs of the single population coacervates with the different 14-3-3 isoforms showed that, indeed, similar recruitment levels of c-Raf pS were observed after overnight incubation (Fig. 2b and Supplementary Fig. 3). Next, we introduced competition for client peptide recruitment by evaluating a 1:1 mixture of 14-3-3σ and 14-3-3γ loaded coacervates (Fig. 2c). After mixing of the two populations of coacervates, c-Raf pS was added at 25 nM, yielding a 14-3-3/c-Raf pS binding site ratio of 2:1. After overnight equilibration, confocal micrographs showed preferential recruitment of c-Raf pS into the coacervates containing 14-3-3γ, the strongest isoform (Fig. 2d and Supplementary Fig. 3). Quantification revealed a statistically significant difference in c-Raf pS recruitment, with a 1.9-fold higher mean recruitment in the 14-3-3γ population compared to the 14-3-3σ population (Fig. 2e, f). The competitive recruitment also demonstrates that although 14-3-3 client recruitment may not be distinguishable by individual analyses of moderately and strongly recruiting coacervates, the mixing of such droplets allows to differentiate these events and determine the degree of competitive binding.

This 1.9-fold enrichment, however, is lower than the expected fold change (34-fold)[37], calculated using a thermodynamic model based on biochemical solution data (Supplementary Fig. 4a, b). This can be explained by the partial exchange of 14-3-3 proteins between the coacervate populations (Supplementary Figs. 4c and 5). This exchange is more prominent for 14-3-3γ than for 14-3-3σ, because the 14-3-3σ dimer is a more stable

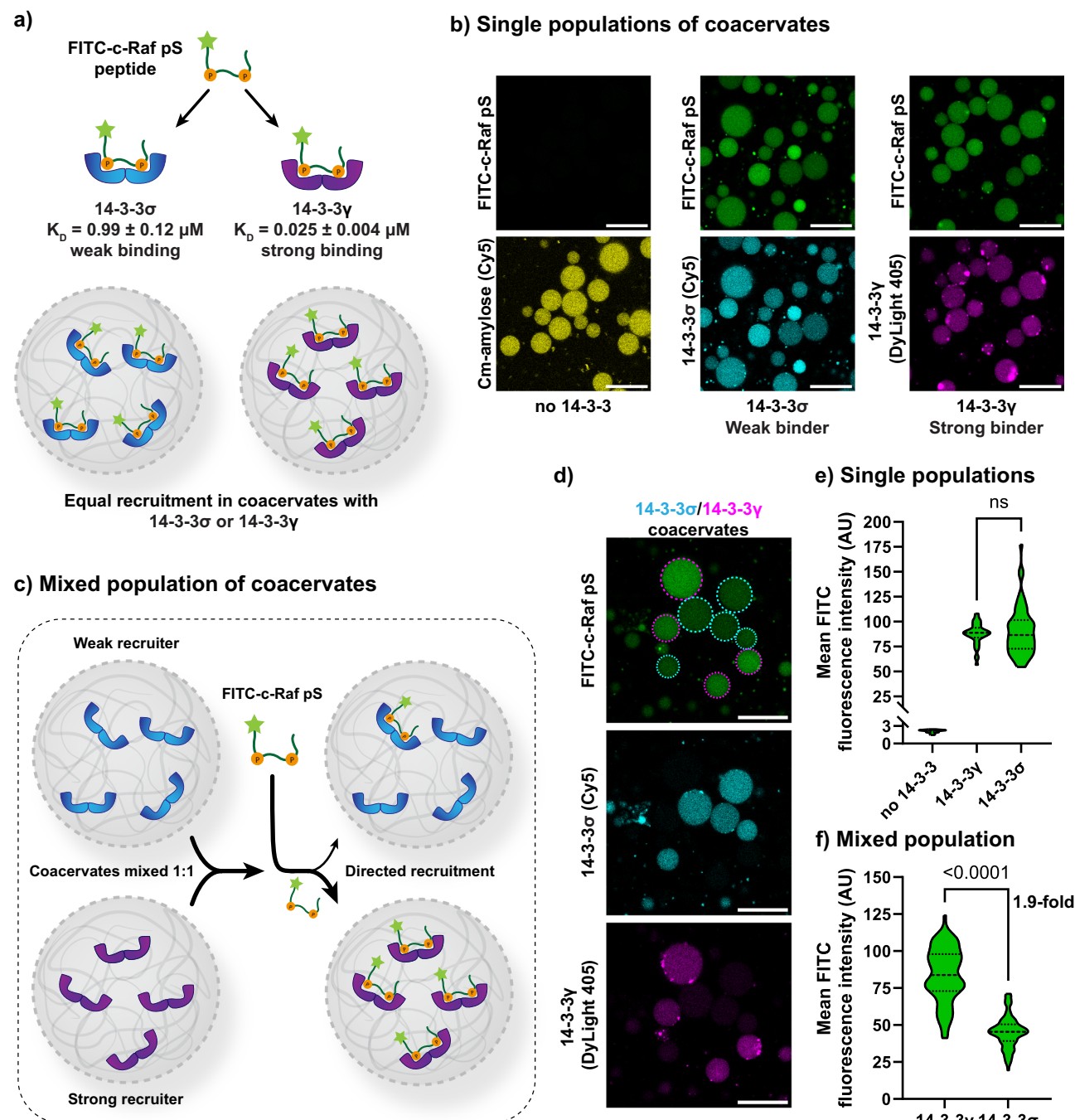

**Fig. 2 | Directed recruitment of the c-Raf phosphopeptide into coacervates with the highest affinity isoform of 14-3-3. a** Schematic overview of binding of the FITC-c-Raf pS peptide to 14-3-3 isoforms and uptake of FITC-c-Raf pS in individual populations of coacervates containing either the weak 14-3-3σ isoform or the strong 14-3-3γ isoform. **b** Confocal micrographs showing the uptake of FITC-c-Raf pS in individual populations in the absence or presence of different 14-3-3 isoforms. Scale bar: 25 μm. Uncropped images are available in Supplementary Fig 3. **c** Schematic overview of competitive uptake of FITC-c-Raf pS into the coacervates containing two 14-3-3 isoforms with different affinity. **d** Confocal micrograph of competitive FITC-c-Raf pS uptake in a mixed coacervate sample containing coacervates loaded with either 14-3-3σ or with 14-3-3γ. Scale bar: 25 μm. Uncropped images are available in Supplementary Fig. 3. **e, f** Quantification of the FITC-c-Raf pS signal from micrographs of individual populations (**e**) or mixed (**f**) coacervate populations containing different 14-3-3 isoforms. Statistical differences were analyzed by non-parametric Dunn's test with correction for multiple comparisons, with $N \geq 21$ coacervates across multiple imaging positions in the same sample. $P$ values are shown above the comparison. Dashed lines represent the median and dotted lines represent the upper and lower quartiles. ns: no statistical difference. The fold difference is given as the fold difference between the means.

complex than 14-3-3γ. 14-3-3σ has 3 salt bridges at its homodimer interface whereas the 14-3-3γ homodimer only has 2 salt bridges[38]. As incorporation of 14-3-3 in the coacervates is governed by interactions of their His-tags with Ni-NTA-Am, the more stable 14-3-3σ dimer is effectively anchored into the coacervates via a double His tag, whereas the more dynamic 14-3-3γ will also be present in its monomeric form carrying only one His tag. Thus, a more dynamic 14-3-3 dimerization process could lead to more dynamic protein exchange between coacervates. This indicates the differentiated recruitment of the c-Raf peptide is governed by the degree of exchange of 14-3-3 between coacervates.

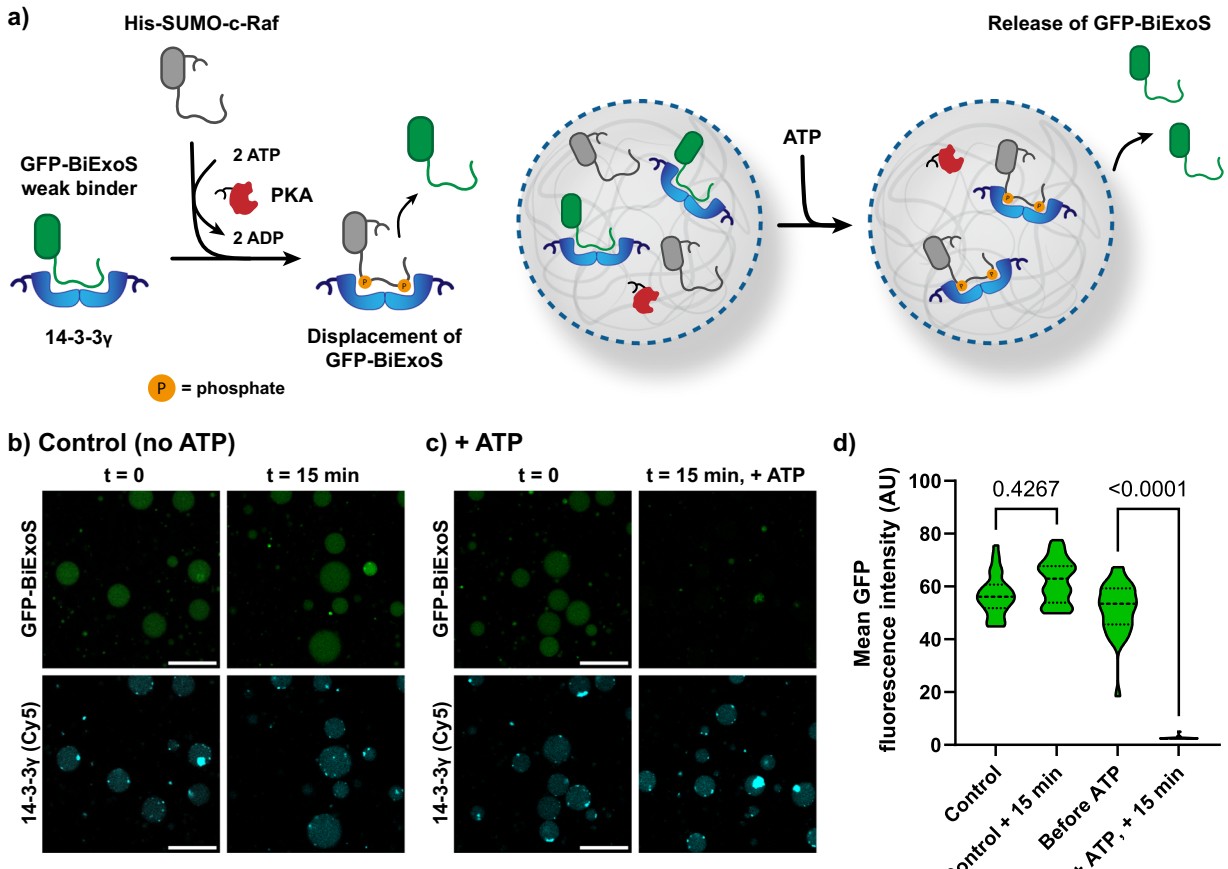

**Fig. 3 | Phosphorylation of His-SUMO-c-Raf drives the release of the client protein GFP-BiExoS by competitive binding to 14-3-3. a** Schematic overview of displacement of GFP-BiExoS by $His_{10}$-SUMO-c-Raf from coacervates after phosphorylation of the c-Raf domain. **b, c** Confocal micrographs of the control sample in the absence of ATP (**b**) and the sample demonstrating triggered release of GFP-BiExoS in the presence of ATP (**c**). Conditions: 100 nM of 14-3-3γ (Cy5-labeled), 10 nM of PKA (His-tagged), 50 nM of $His_{10}$-SUMO-c-Raf, 50 nM of GFP-BiExoS, bulk concentrations. Uncropped images are available in Supplementary Fig. 11.

Scale bar: 25 μm. **d** Quantification of the GFP-BiExoS signal in micrographs in panels **b** and **c**. Statistical differences were analyzed by nonparametric Dunn's test with correction for multiple comparisons, with $N \geq 31$ coacervates across multiple imaging positions in the same sample. *P* values are shown above the comparison. Dashed lines represent the median and dotted lines represent the upper and lower quartiles.

## Intracellular competition of 14-3-3 binding partners via affinity regulation

We subsequently investigated intracellular competition of binding partners for 14-3-3 (Fig. 3a) via affinity regulation. Competition was achieved by loading two binding partners in the same coacervate, namely the bivalent phosphorylation-independent 14-3-3 interaction domain BiExoS L423A (BiExoS) and the bivalent phosphorylation-dependent c-Raf peptide domain. The BiExoS domain, which is derived from the native 14-3-3-binding domain of the bacterial toxin Exoenzyme S[39], has moderate affinity for 14-3-3 (Supplementary Fig. 6). To visualize its presence, it was fused to green fluorescent protein (GFP). The c-Raf peptide domain was fused to a His-SUMO domain, where the SUMO-tag serves as a solubility tag enhancing protein expression, yielding the His-SUMO-c-Raf fusion protein. In the unphosphorylated state this c-Raf domain has no affinity for 14-3-3 and is outcompeted by GFP-BiExoS for its binding to 14-3-3. However, as already shown in Fig. 2, the phosphorylated c-Raf sequence has strong affinity for 14-3-3. Double phosphorylation by protein kinase A (PKA) of the c-Raf domain therefore provides a powerful affinity switch of this protein construct for 14-3-3 (Supplementary Fig. 7).

Coacervates were prepared with 14-3-3γ (100 nM), His-SUMO-c-Raf (50 nM), and His-tagged PKA (10 nM). GFP-BiExoS (50 nM) was efficiently sequestered in the coacervates in a 14-3-3-dependent manner after equilibration for >5 h, as analyzed by confocal microscopy (Supplementary Figs. 8 and 9). Since both BiExoS and c-Raf domains bind in a bivalent manner and to ensure efficient competition for 14-3-3 binding, 14-3-3 was loaded in a twofold molar excess, yielding a final ratio of 14-3-3/GFP-BiExoS/His-SUMO-c-Raf of 1:1:1 with respect to binding sites. In the control sample without the PKA substrate ATP, GFP-BiExoS remained in the coacervates over time as analyzed by confocal microscopy (Fig. 3b). Upon the addition of ATP, rapid release of GFP-BiExoS from the coacervates was observed due to phosphorylation of the competing His-SUMO-c-Raf client and subsequent displacement of GFP-BiExoS (Fig. 3c, d). GFP-BiExoS was efficiently released during an equilibration time of 15 min. We confirmed that this effect was mediated by kinase activity by performing a control experiment in the absence of kinase, which showed no ATP-dependent effect (Supplementary Fig. 10). This demonstrates that intracellular affinity modulation can control the orthogonality of PPIs, which leads to displacement of a client protein in an enzymatically regulated manner.

## Engineering competitive protein recruitment between artificial cells

Next, we aimed to couple this intracellular competition process with a controlled transfer of a protein binding partner between two coacervates (Fig. 4a). GFP-BiExoS, after its release from the 'sender' coacervates by competitive binding of the phosphorylated His-SUMO-c-Raf protein, can be taken up by a 'receiver' population of coacervates, which also contain 14-3-3. Since the

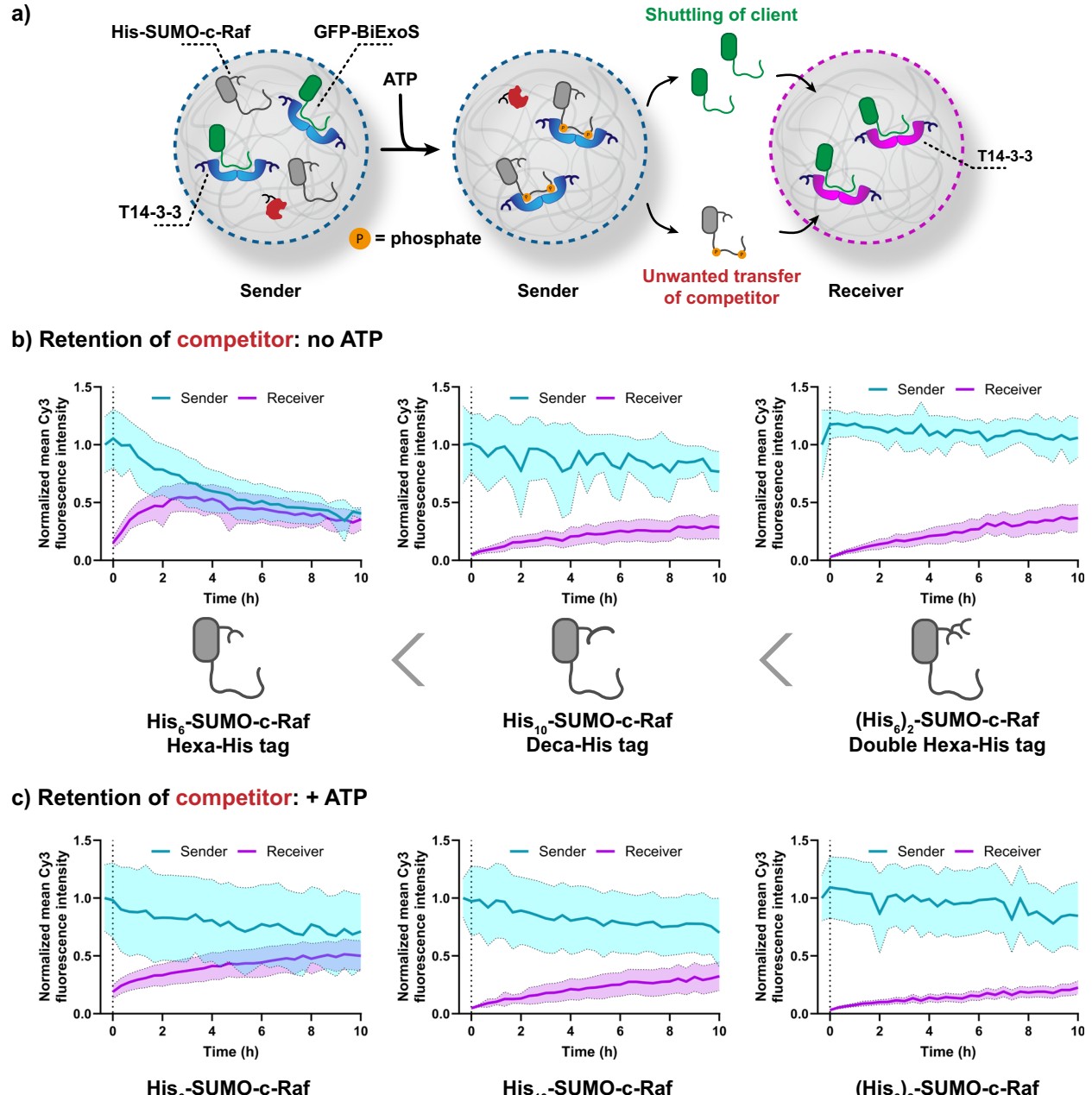

**Fig. 4 | His-tag affinity engineering of competitor His-SUMO-c-Raf promotes its retention in coacervates. a** Schematic overview of displacement of GFP-BiExoS by SUMO-c-Raf from coacervates after phosphorylation of the c-Raf domain. The client protein GFP-BiExoS can be taken up by the receiver population of coacervates, which also contain 14-3-3. **b, c** Quantification of confocal micrographs demonstrating the transfer from the sender to the receiver coacervates of SUMO-c-Raf variants (Cy3-labeled) in the absence (**b**) or presence (**c**) of ATP (1 mM). The data is shown as mean ± standard deviation. Sender coacervates were imaged, after which the receiver population (sender/receiver ratio: 1:1 by volume) and ATP were added. The Cy3 intensity is normalized to the initial timepoint in all cases. Conditions: sender coacervates were loaded with 100 nM of T14-3-3 (Cy5-labeled), 10 nM of PKA (His-tagged), 50 nM of His-SUMO-c-Raf variant, and 50 nM of GFP-BiExoS, bulk concentrations. Receiver coacervates were loaded with 100 nM of T14-3-3 (DyLight 405-labeled), bulk concentration. Images are available in Supplementary Figs. 12–14, and an additional comparison of GFP-BiExoS shuttling between the samples is available in Supplementary Fig. 15.

His-SUMO-c-Raf protein competes with GFP-BiExoS for 14-3-3 binding sites, it is essential that it is kept in the sender population by its His-tag and does not partition in the receiver population. Hence, we engineered three variants of the protein: His$_6$-SUMO-c-Raf, with a hexahistidine tag, His$_{10}$-SUMO-c-Raf, with a 10-mer His-tag, and (His$_6$)$_2$-SUMO-c-Raf, with a tandem 6-mer His-tag separated by a short (GGS)$_2$ spacer.

We first investigated the retention of the engineered His-SUMO-c-Raf variants (Cy3-labeled) in the sender coacervates; the moderate affinity T14-3-3 isoform (100 nM), His-SUMO-c-Raf (25 nM), and His-tagged PKA (catalytic subunit, 10 nM) were added, whereas the 'receiver' coacervates

were loaded with T14-3-3 (100 nM) only. T14-3-3 was chosen as scaffold protein to balance binding and release features. The sender and receiver populations were prepared separately, and the client protein GFP-BiExoS was added solely to the sender coacervates. After equilibration for >5 h to allow for protein uptake, the samples were mixed, analyzed by confocal microscopy, and quantified (Fig. 4b, c and Supplementary Figs. 12–14). His$_6$-SUMO-c-Raf, in the absence of ATP, quickly divided over both sender and receiver populations based on its weak affinity for the Ni-NTA-amylose (present in both populations). His$_{10}$-SUMO-c-Raf was found to be retained in the sender coacervates in a more stable way owing to its higher affinity for

Ni-NTA-Am. The highest affinity variant, (His)$_2$-SUMO-c-Raf with its tandem 6-mer His-tag, performed similarly to His$_{10}$-SUMO-c-Raf, with 26% and 27% relative fluorescent signal of the His-SUMO-c-Raf protein in the receivers at the 10 h timepoint, respectively (Fig. 4b). The same trend was observed in the presence of ATP, with the general difference that the phosphorylated c-Raf proteins were retained in the sender coacervates to a higher degree owing to additional interactions with 14-3-3.

After establishing optimal conditions under which the c-Raf peptide was retained in the sender population, the displacement and transfer of the client GFP-BiExoS from the senders could be studied by c-Raf phosphorylation. However, since both the sender and receiver coacervates were loaded with the equal 14-3-3 isoform and concentration, GFP-BiExoS was already translocated to the receivers in an untriggered manner (Supplementary Figs. 12–14). Hence, the GFP-BiExoS recruitment in the receiver coacervates mediated by intracellular competition in the senders did not differ significantly from the samples without ATP.

As 14-3-3 isoforms were found to direct the c-Raf peptide uptake (Fig. 2), we opted to use the difference in 14-3-3 isoform affinity to regulate the transfer of GFP-BiExoS from the sender to the receiver population. 14-3-3γ, the highest affinity isoform, was loaded in the sender population to initially capture GFP-BiExoS more strongly with a $K_D$ of 59 ± 4 nM (Supplementary Fig. 6). The receiver coacervates contained the lowest affinity isoform, 14-3-3σ, to provide a weaker background recruitment of GFP-BiExoS (Fig. 5a), with a $K_D$ of 23 ± 1 μM (Supplementary Fig. 6). To enable subsequent efficient directed uptake of GFP-BiExoS, a higher concentration of 500 nM of 14-3-3σ was loaded in the receiver population, compared to 100 nM of 14-3-3γ in the senders.

The sender and receiver population of coacervates were mixed in a 1:1 ratio and analyzed by confocal microscopy over time. In the absence of ATP, non-triggered transfer of GFP-BiExoS to the receiver population was still observed (Fig. 5b, c), but to a lower degree than when equal isoforms were loaded in both populations. Upon the addition of ATP, GFP-BiExoS was efficiently released from the sender population due to the phosphorylation of the competing client protein His$_{10}$-SUMO-c-Raf by PKA (Fig. 5d, e). GFP-BiExoS, following its release, was taken up in the receiver population. Here, the programmed shuttling mediated by phosphorylation was significantly different from the untriggered recruitment of GFP-BiExoS in the receiver population (Fig. 5f). The 14-3-3 isoform-dependent shuttling demonstrates that differences in the affinity of a client protein to a hub protein embedded in the coacervate can be applied to effectively transport the client protein between populations.

## Discussion

Signaling between cells by exchange of proteins is one of the mechanisms of intercellular communication, often mediated by specific PPIs. Here, we demonstrated the use of the 14-3-3 scaffold protein to differentiate the recruitment of its interaction partners in specific populations of coacervate-based artificial cells by affinity modulation.

Recruitment of the c-Raf peptide was observed to be equal in coacervates loaded with either 14-3-3γ, the strongest binding isoform, or 14-3-3σ, the weakest binding isoform. However, when the two coacervate populations were mixed and the peptide was added, differentiated recruitment of the c-Raf peptide into the 14-3-3γ coacervates was observed. This demonstrates that recruitment in such phase separated droplets is driven by affinity when there is an excess of recruiting proteins, but that differentiated uptake can still take place in mixed populations. Intracellular competitive PPIs were used to displace the moderately binding client GFP-BiExoS from 14-3-3 upon phosphorylation of the competing client His-SUMO-c-Raf by kinase PKA. This highlights that competitive PPIs can be used to regulate the protein composition in coacervates.

Finally, we coupled intracellular competition and protein release to a directed transfer of a protein-binding partner from a sender to a receiver population. As the natural partitioning of GFP-BiExoS between coacervates containing the same 14-3-3 isoform prevented significant active translocation of GFP-BiExoS, affinity regulation had to be introduced in both the

binding partners and the different 14-3-3 scaffolds in the sender and receiver populations. By engineering the system to have the strongest 14-3-3 isoform in the sender population and a fivefold excess of the weakest 14-3-3 isoform in the receiver population, significant directed translocation of GFP-BiExoS was observed. This demonstrated that the affinity-regulated exchange between coacervates can overcome the natural partitioning behavior of the dynamic macromolecular cargo.

The results here demonstrate that PPIs can be used to direct protein clients to distinct coacervates. After demonstrating efficient protein translocation between coacervates, we envision the transport of an active protein for a synthetic signaling pathway, such as a kinase. We also envision the substitution of 14-3-3 with other hub proteins such as PSD-95, which is involved in neuronal cell-cell signaling. Such systems could be used to better understand biological signaling, where differences in affinity and competing binders could influence the uptake of proteins in distinct cells or in distinct intracellular compartments.

## Methods
### Materials and instruments
His-tagged full-length 14-3-3σ and 14-3-3γ were kindly provided by Maxime van den Oetelaar and Marloes Pennings. The pOPINF plasmid containing T14-3-3-cΔc was kindly provided by Dr Anniek den Hamer. The FITC-c-Raf pS233/pS259 peptide was a kind gift from Emira Visser. The (unphosphorylated) FITC-c-Raf S233/S259 peptide was a kind gift from Lenne Lemmens. The BiExoS L423A peptide was kindly provided by Dr Stijn Aper. The full chemical identity of peptides is given in Supplementary Table 1. $^1$H NMR spectra were collected on an AVANCE III HD (400 MHz) NMR spectrometer (Bruker). The $^1$H NMR chemical shift values are reported in ppm relative to the residual solvent peak.

### DNA molecular biology and cloning
The sequence of PKA (catalytic subunit only) was derived from UniProt ID P17612 (cAMP-dependent protein kinase catalytic subunit alpha). The sequence of superfolder GFP was based on work by Pédelacq et al.[40]. Protein sequences and physicochemical properties are given in Supplementary Table 2. All DNA was ordered through Integrated DNA Technologies (IDT). The constructs were codon-optimized using IDT's built-in codon optimization tool for *Escherichia coli* (*E. coli*). The pET28a vector and gBlock dsDNA fragments were digested with the appropriate restriction enzymes (New England Biolabs). After ligation into the vector, the constructs were verified using Sanger sequencing (Azenta). Constructs were transformed into BL21(DE3) *E. Coli* cells (Novagen).

### Expression of T14-3-3 and His-SUMO-c-Raf variants
For T14-3-3 and His-SUMO-c-Raf S233/S259 variants, 1L or 0.5L of 2xYT medium supplemented with the appropriate antibiotic was used, which was in the case of T14-3-3-His, ampicillin (100 μg mL$^{-1}$) and for the other constructs 30 μg mL$^{-1}$ of kanamycin. After inoculation using an overnight culture grown at 37 °C, 250 rpm, the culture was grown to an optical density (OD600) of 0.6 at 37 °C, 140 rpm. Then, protein expression was induced by the addition of isopropyl β-D-1-thiogalactopyranoside (IPTG) at a final concentration of 0.5 mM, with incubation overnight at 18 or 20 °C, 140 rpm. Cells were harvested by centrifugation at 4 °C and 10,000x$g$ for 15 minutes. The cell pellets were resuspended in lysis buffer (50 mM Tris, 300 mM NaCl, 30 mM imidazole, pH 8.0). Cells were lysed using an EmulsiFlexC3 High-Pressure homogenizer (Avestin) at 15,000 psi for three consecutive rounds. Cell debris and insoluble proteins were removed by centrifugation at 4 °C and 35000x$g$ for 20 minutes. His-tagged proteins were purified from the soluble lysate using Ni-NTA affinity chromatography (His-Bind Resin, Novagen). The lysate was loaded onto the His-bind resin and washed twice with wash buffer (50 mM Tris, 300 mM NaCl, 60 mM imidazole, pH 8.0). The His-tagged proteins were eluted from the resin using elution buffer (50 mM Tris, 300 mM NaCl, 250 mM imidazole, pH 8.0). The eluted fractions were analyzed using SDS-PAGE (4–20% Mini-PROTEAN TGX Precast Protein Gel, Bio-Rad) and the purest fractions were pooled.

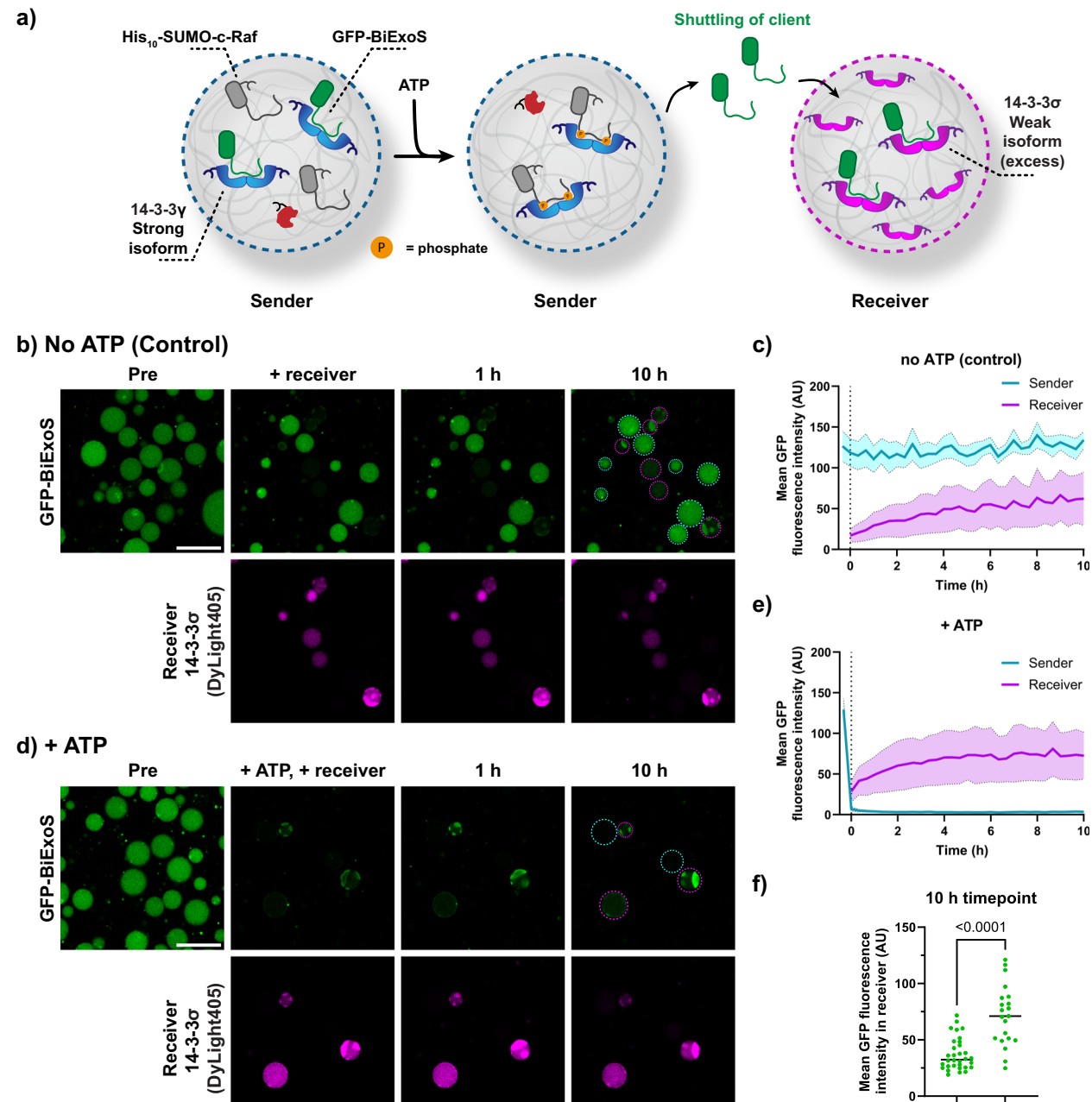

**Fig. 5 | Improved shuttling of GFP-BiExoS mediated by 14-3-3 isoform differences. a** Schematic overview of displacement of GFP-BiExoS by SUMO-c-Raf from coacervates after phosphorylation of the c-Raf domain. The client protein GFP-BiExoS is taken up by the receiver population of coacervates, which contain a fivefold excess of the weak 14-3-3σ isoform. **b–e** Confocal micrographs of the control sample in the absence of ATP (**b**) and the sample demonstrating triggered signaling of GFP-BiExoS in the presence of ATP (**d**), and quantification of the GFP-BiExoS signal from the micrographs (**c, e**) with data shown as mean ± standard deviation. Conditions: Senders were loaded with 100 nM of 14-3-3γ (Cy5-labeled), 10 nM of PKA (His-tagged), 25 nM of His$_{10}$-SUMO-c-Raf, 50 nM of GFP-BiExoS, bulk concentrations.

Receiver coacervates were loaded with 500 nM (bulk concentration) of 14-3-3σ (DyLight 405-labeled). Uncropped images are available in Supplementary Fig. 16. Colored outlines of sender and receiver coacervates were added as a visual guide. Scale bar: 25 μm. **f** Quantification of micrographs taken at distinct positions at the 10 h timepoint, showing the GFP signal in the receiver population of coacervates. Statistical differences were analyzed by nonparametric Dunn's test with correction for multiple comparisons, with $N \geq 21$ coacervates across multiple imaging positions in the same sample. $P$ values are shown above the comparison. Dashed lines represent the median and dotted lines represent the upper and lower quartiles.

The combined fractions were extensively dialyzed against coacervate buffer (20 mM HEPES, 100 mM KCl, pH 7.5, freshly prepared with 100 μM tris(2-carboxyethyl)phosphine (TCEP) using membrane tubing with a molecular weight cut-off (MWCO) of 12–14 kDa (Fisher Scientific). Protein concentration was determined using an ND-1000 spectrophotometer (Thermo Scientific) at 280 nm with theoretical extinction coefficients as determined by the Expasy ProtParam tool as shown in Table S2, in the case of T14-3-3. In the case of the His-SUMO-c-Raf S233/S259 variants, the extinction

coefficient was low due to the absence of Trp residues. For these three constructs, a bicinchoninic acid (BCA) assay was used (Thermo Fisher Pierce™ BCA Protein Assay Kit) for determination of protein concentration according to the manufacturer's protocol. The purified proteins were aliquoted, flash-frozen in liquid N$_2$, and stored at −80 °C for single-use aliquots. The identity and purity of the protein samples were confirmed using liquid chromatography quadrupole time of flight mass spectrometry (LC–MS Q-ToF) in Supplementary Figs. 17–20.

## Expression of PKA-His

2L of TB auto-induction medium supplemented with kanamycin (30 µg mL$^{-1}$) was inoculated using an overnight culture grown at 37 °C, 250 rpm. The culture was grown at 37 °C, 140 rpm, for 4 h, after which protein expression was carried out overnight at 25 °C, 140 rpm. Cells were harvested by centrifugation at 4 °C and 10,000 x$g$ for 15 minutes. The cell pellet was resuspended in lysis buffer (50 mM Tris, pH 8, 600 mM NaCl, 30 mM imidazole, 5% glycerol, pH 8.0). Cells were lysed using an Emulsi-FlexC3 High-Pressure homogenizer (Avestin) at 15,000 psi for three consecutive rounds. Cell debris was removed by centrifugation at 4 °C and 35000 xg for 20 minutes. PKA-His was purified from the soluble lysate using Ni-NTA affinity chromatography (His-Bind Resin, Novagen). The lysate was loaded onto the His-bind resin and washed twice with lysis buffer (50 mM Tris, pH 8, 600 mM NaCl, 30 mM imidazole, 5% glycerol, pH 8.0). The His-tagged proteins were eluted from the resin using elution buffer (50 mM Tris, pH 8, 600 mM NaCl, 250 mM imidazole, 5% glycerol, pH 8.0). Subsequently, the fractions were loaded on a pre-equilibrated 2 mL Strep-Tactin XT column (Iba Lifesciences). After 2 repeats of 5 column volumes of washing with wash buffer (100 mM Tris pH 8, 150 mM NaCl, 1 mM EDTA), the protein was eluted using freshly prepared Strep elution buffer (100 mM Tris pH 8, 150 mM NaCl, 1 mM EDTA, 50 mM biotin). The eluted fractions were analyzed using SDS-PAGE (4–20% Mini-PROTEAN TGX Precast Protein Gel, Bio-Rad) and the purest fractions were pooled. The protein was extensively dialyzed against storage buffer (50 mM HEPES, 100 mM KCl, pH 7.5) using membrane tubing with a MWCO of 12–14 kDa (Fisher Scientific). Protein concentration was determined using an ND-1000 spectrophotometer (Thermo Scientific) at 280 nm with a theoretical extinction coefficient of 59270 M$^{-1}$ cm$^{-1}$ as determined by the Expasy ProtParam tool. The protein was aliquoted into single-use aliquots, flash-frozen in liquid N$_2$, and stored at −80 °C. The identity and purity of the protein sample were confirmed using LC–MS Q-ToF (Supplementary Fig. 21).

## Expression of GFP-BiExoS L423A

1L of LB medium supplemented with kanamycin (30 µg mL$^{-1}$) was inoculated using an overnight culture grown at 37 °C, 250 rpm. The culture was grown to an OD600 of 0.6, and then induced by the addition of IPTG (0.5 mM final concentration). Protein expression was carried out overnight at 20 °C, 150 rpm. Cells were harvested by centrifugation at 4 °C and 10,000x$g$ for 15 minutes. The cell pellet was resuspended in lysis buffer (100 mM Tris, 150 mM NaCl, 1 mM EDTA, pH 8.0). Cell lysis was performed by ultrasonic disruption in on/off cycles of 5 s/10 s respectively for a total of 10 min (70% amplitude, Branson Sonifier 150). Cell debris and insoluble proteins were removed by centrifugation at 4 °C and 20,000 x$g$ for 20 minutes. The soluble lysate was applied to a Strep-Tactin gravity flow column (Strep-Tactin®XT 4Flow® resin, IBA Lifesciences) that was equilibrated with the lysis buffer. The resin was washed twice with lysis buffer. The protein was eluted from the resin using elution buffer (100 mM Tris/HCl, 150 mM NaCl, 1 mM EDTA, 50 mM biotin, pH 8.0). The eluted fractions were analyzed using SDS-PAGE (4–20% Mini-PROTEAN TGX Precast Protein Gel, Bio-Rad) and the purest fractions were pooled. The protein was extensively dialyzed against coacervate buffer (20 mM HEPES, 100 mM KCl, pH 7.5, with freshly added 100 µM of TCEP) using membrane tubing with a MWCO of 12–14 kDa (Fisher Scientific. Protein concentration was determined using an ND-1000 spectrophotometer (Thermo Scientific) at 280 nm with a theoretical extinction coefficient as determined by the Expasy ProtParam tool (Table S2). The protein was aliquoted into single-use aliquots, flash-frozen in liquid N$_2$, and stored at −80 °C. The identity and purity of the protein sample were confirmed using LC-MS Q-ToF (Supplementary Fig. 22).

## LC-MS Q-ToF

The mass and purity of the proteins were determined using a high-resolution LC-MS Q-ToF system consisting of an ACQUITY UPLC I-Class system (Waters) coupled to a Xevo G2 quadrupole time of flight. The protein was separated (0.3 mL min$^{-1}$) on a column (Polaris C18A reverse phase column 2.0 × 100 mm, Agilent) using a 15–75% acetonitrile gradient in water supplemented with 0.1% v/v formic acid before analysis in positive mode in the mass spectrometer. The m/z spectra were deconvoluted using the MaxENTI algorithm in the Masslynx v4.1 software.

## Labeling of 14-3-3 proteins with fluorescent dyes

For aspecific fluorescent labeling of 14-3-3, dyes with NHS ester reactivity were used. DyLight 405 NHS ester (Thermo Fisher) and sulfo-Cy5 NHS ester (Lumiprobe) were dissolved at 10 mg/mL in DMSO. Proteins were diluted at least tenfold with labeling buffer (0.1 M NaHCO$_3$, pH 8.5). 1.5x to 3.0x of molar excess of dye relative to the protein was added to the protein solution and the mixture was incubated at 4 °C for 3 hours. Unreacted dye was removed twofold using PD Minitrap G-25 size exclusion column, equilibrated with coacervate buffer (20 mM HEPES, 100 mM KCl, pH 7.5). The average labeling per protein was measured using the absorption of the dye at their maximum absorption wavelength and at 280 nm for the protein, using the manufacturer-provided extinction coefficients and A$_{280}$ correction factors for the dyes.

## Synthesis of amylose derivatives and terpolymer

The procedures for polymer synthesis are available in the Supplementary Methods in the Supplementary Information.

## Coacervate preparation

Q-Am, Cm-Am, and NTA-Am were dissolved separately in coacervate buffer (20 mM HEPES, 100 mM KCl, pH 7.5) at a concentration of 1 mg mL$^{-1}$. First, buffer and NTA-Am were added to 7.5 µM of NiCl$_2$ (final concentration) in a 1.5 mL tube shaking at 1500 rpm in a MixMate shaker (Eppendorf). Consecutively, Cm-Am and Q-Am were added to induce coacervation in a 1.7:0.8:0.2 mass ratio of Q-Am:Cm-Am:NTA-Am, corresponding to a 2.5:0.8:0.2 charge ratio due to differing degrees of substitution, which was found to be the most stable. After 30 s, His-tagged protein cargo was added to the shaking solution. To achieve stabilized particles, 3.3 µL terpolymer (50 mg mL$^{-1}$ in methoxy-poly(ethylene glycol) 350 on coacervate volume of 100 µL) was added after 6 min and the mixture was shaken for another 5–10 s. For the communication experiments, unencapsulated His-tagged proteins were removed by a centrifugation protocol: the coacervate samples were centrifuged at 250 g for 4 minutes, after which 80% of the supernatant was removed and replenished with coacervate buffer. For microscopy, 50–100 µL of each sample was loaded on a µ-slide 18 well glass bottom (Ibidi). In the case of experiments involving PKA, MgCl$_2$ was supplemented (5 mM final concentration).

## Fluorescence anisotropy assays

14-3-3 was titrated in a 2-fold dilution series to 10 nM of fluorescently labeled peptide in coacervate buffer (20 mM HEPES, 100 mM KCl, pH 7.5) supplemented with 0.1% (v/v) of Tween 20 and 1 mg/mL of bovine serum albumin (BSA) to prevent aspecific hydrophobic interactions. Dilution series were prepared in low volume, non-binding polystyrene 384-well plates (Corning 4514 Black Round Bottom 384-well plates). Measurements were performed directly after plate preparation using a Tecan Spark plate reader at room temperature. The following settings were used: excitation 485 ± 20 nm; emission: 535 ± 25 nm; mirror: Dichroic 510; number of flashes: 30; integration time: 40 µs; settle time: 1 ms; gain: optimal; and Z-position: calculated from well. Wells containing only the labeled peptide were used to set as G-factor at 35 mAU. All data were analyzed using GraphPad Prism (version 10.0.3) and fitted using a four-parameter logistic model (4PL) to determine binding affinities ($K_D$). All results are based on triplicates, with the mean and standard error determined by GraphPad.

## Confocal laser scanning microscopy

Confocal laser scanning microscopy (Leica TCS SP8) was used for analysis of coacervates with fluorescent cargo. The system was equipped with a 405 nm laser (used for DyLight 405), a 488 nm laser (used for FITC and

GFP), 552 nm laser (used for Cy3), and 638 nm laser (used for Cy5) and a hybrid detector (HyD). Upon excitation with the 405 nm laser, emission was collected between 415 and 470 nm. For the 488 nm laser channel, emission was collected between 498 and 550 nm. For the 552 laser channel, emission was collected between 562 and 630 nm. Finally, for the 638 laser channel, emission was collected between 648 and 710 nm. Laser power and detector gain were optimized for each different construct and concentration to use the maximum number of gray values of the detector. For single timepoint measurements, an HC PL APO CS2 63 × water immersion objective with a numerical aperture (NA) of 1.20 was used. Images were acquired with a resolution of 1024 × 1024 pixels and a pixel dwell time of 1.2 µs. For kinetic measurements, an HC PL FLUOTAR × 63 dry objective with an NA of 0.90 was used. Images were acquired with a resolution of 1024 × 1024 and a pixel dwell time of 600 ns at timepoints on certain positions using the Mark and Find tool. The pinhole was set to 1 Airy Unit for the wavelength of maximum emission for each fluorophore.

### Image processing and analysis

All images were processed and analyzed with Fiji (ImageJ). For micrographs with 14-3-3, the brightness was digitally adjusted equally for enhanced visibility. The channels of the client were not adjusted and uncropped images in the SI were not adjusted either. For quantification of the internal fluorescence intensity, a threshold was applied to images in the 14-3-3 channel, converting it into a binary image. Next, the images were dilated using a maximum filter of radius 1 pixel, to make particle outlines more pronounced and particle recognition more reliable. Next, a watershed function was applied to separate adjacent coacervates into individual regions of interest (ROIs). Using the particle analysis tool with appropriate cutoff values to select coacervates as ROIs, fluorescence intensity was quantified. The ROIs recognized in the 14-3-3 channel were also redirected to the client peptide or protein channels, which is especially important for experiments over time since the 14-3-3 channel is relatively constant in fluorescence. Alternatively, for images without 14-3-3, Cy5-labeled Cm-Am was used to determine the coacervate outlines. ROIs were visually checked to make sure that only coacervates were selected. Next, recognized ROIs were filled and measured using the particle analysis tool, redirected to the peptide channel. The intensity was determined for each selected particle.

### Reporting summary

Further information on research design is available in the Nature Portfolio Reporting Summary linked to this article.

### Data availability

All data supporting the findings of this study are available within this article, the Supplementary Information, and the Supplementary Data. Supplementary Data 1 contains the NMR spectra. Supplementary Data 2 contains the source data for the main figures. Additional data related to this study are available from the corresponding author upon reasonable request.

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

## Acknowledgements

The Dutch Ministry of Education, Culture, and Science (Gravitation Program 024.001.035 and Growth Fund "Big Chemistry") is acknowledged for funding. Joost van Dongen and Sebastian van den Wildenberg are thanked for Q-ToF LC-MS measurements. Yannick Leurs is thanked for fruitful discussions regarding the His-SUMO-c-Raf proteins. Maarten Merkx is acknowledged for providing feedback on the manuscript.

## Author contributions

T.W.V. designed and performed the experiments and analyzed the experimental results. S.N. performed synthesis of the terpolymer and provided input on coacervate formulations. M.A.M.V. performed initial experiments. L.B. and J.C.M.H. conceived and supervised the project. T.W.V. wrote the manuscript with the aid and input of all other authors.

## Competing interests

The authors declare the following conflict of interest: L.B. is a co-founder and shareholder of Ambagon Therapeutics. The remaining authors declare no competing interests.
