## [Peer Review File · Communications Chemistry]

Reviewers' comments:

Reviewer #1 (Remarks to the Author):

In this interesting manuscript the authors characterize competitive binding and communication between coacervates utilizing 14-3-3 binding proteins and ligands of differing affinity. Their complex coacervates, also termed 'artificial cells' by the authors, are formed by association of positively charged quaternized amylose and negatively charged carboxymethylated amylose. They incorporate Ni-NTA to tether his-tagged proteins and anchor 14-3-3 protein isoforms, which have distinct affinities for substrates, for example ranging from 25-1000 nM for the c-Raf pS peptide. They demonstrate selective recruitment of target peptides dependent on presence of 14-3-3 domain in the coacervates and demonstrate competitive recruitment in which the coacervates containing the higher affinity binding isoform of 14-3-3 better recruit a FITC-labelled c-Raf pS peptide. Intriguingly, target phosphorylation can selectively effect its binding, causing release from the coacervates. And they set up a shuttling system to either retain or transfer peptide from sender to receiver coacervates. Strengths of the work include demonstration of the validity of conceptual and experimental framework to create peptide competition and communication between compartments in protocell like systems, and demonstration of fast ATP dependent release of target peptides from coacervates. Enthusiasm is somewhat diminished by the extent of competition and from missing controls.

Major Comment

1. Limited extent of competition and missing controls.

In Fig 2F, despite a 40-fold difference in peptide affinity, the authors achieve only ~ 1.5-2-fold enrichment of target in the coacervate containing the strong binding domain. Can they not optimize the reaction conditions and protein concentration to achieve a higher differential partitioning?

Missing controls. ATP is known to act as a hydrotrope and thus for Fig 3D, the proper control is not just no ATP, but rather removal or the phosphorylation site from the c-Raf peptide. And additionally consider changing it to Glu or Asp as a phosphomimic

Reviewer #2 (Remarks to the Author):

Jan C. M. van Hest and colleagues investigate the control of protein recruitment in artificial cells, with an emphasis on protein exchange across different populations of coacervate-based cells. The work is beautifully written, and the science is sound.

Recommendation: Accept with minor revisions highlighted.

Although the study has some interesting refinements, such as affinity-regulated exchange, competitive binding, intracellular affinity modulation, directed protein transfer, and application of affinity differences, there is very little novelty given that the authors published an article with the exact premise but using different client peptides (Enzymatic Regulation of Protein-Protein Interactions in Artificial Cells, please refer to <https://doi.org/10.1002/adma.202300947>). The novelty of the work has therefore been diluted.

Other minor concerns that need to be addressed by the authors

In Figure S1, the estimation of protein concentration using fluorescence may be inaccurate. The authors can compare fluorescence-based estimation with other spectrometric methods, like the BSA assay.

In Fig. 3d, please specify the fluorescence being quantified on the Y axis. In this case, GFP.

Reviewer #3 (Remarks to the Author):

I co-reviewed this manuscript with one of the reviewers who provided the listed reports. This is part of the Communications Chemistry initiative to facilitate training in peer review and to provide appropriate recognition for Early Career Researchers who co-review manuscripts.

Reviewer #4 (Remarks to the Author):

The manuscript by Veldhuisen et al. elucidates the competitive recruitment of a client peptide into the 14-3-3 protein hub within artificial cells. They employ an affinity-regulated exchange method to demonstrate protein-protein interactions (PPIs) involving the 14-3-3 hub protein within diverse populations of coacervate-based artificial cells. The significance of 14-3-3 proteins in living cell signal transduction arises from their multifaceted roles as molecular scaffolds, regulators of enzyme activity, and modulators of protein localization. This versatility enables precise control of cellular signaling pathways and responses to environmental cues. Leveraging the adaptable binding landscape of 14-3-3 scaffold proteins with their binding partners presents an intriguing opportunity to implement affinity-regulated protein exchange and investigate PPIs in coacervate-based artificial cells. By incorporating different isoforms of 14-3-3 with varying PPI affinities, the system can direct a client peptide to artificial cells with higher binding affinities. Phosphorylation is utilized to manipulate the affinity of client proteins, displacing weaker binding partners from 14-3-3 and releasing them from coacervates. Through affinity engineering and competitive binding, a communication system is established between coacervates, facilitating the transport of client proteins from strongly recruiting artificial cells to those with weaker recruitment capabilities. Demonstrating efficient protein translocation between coacervates suggests the potential for transporting active proteins for synthetic signaling pathways, such as kinases. This system holds promise for advancing our understanding of biological systems, where variations in affinity and competition among binders could influence protein uptake in distinct cells or intracellular compartments.

The utilization of 14-3-3 protein and affinity-based interactions plays a critical role in the dynamics of signal transduction within living cell membranes. This intricate mechanism of protein recruitment not only facilitates targeted uptake but also enables the exchange of client proteins, essential for cellular communication. Replicating this process in artificial cells holds immense promise, offering a means to mimic the sophisticated intercellular chemical communication observed in biological systems. By engineering artificial cells capable of selectively taking up and exchanging proteins, novel avenues for studying intercellular communication and developing biomimetic systems emerge. The ability to engineer coacervates with differentiated behaviors and establish communication systems between them could open up new avenues for studying intercellular communication and developing advanced biomimetic systems for various applications. Therefore, these claims are likely to be of interest to researchers and professionals in the relevant scientific communities, especially in bottom-up synthetic biology and biotechnology field. Conceptually, the idea in the manuscript is straightforward and reasonably laid out. Therefore, I recommend this article for publication in communication chemistry journal.

However, I have few minor questions for the authors:

1. Is it feasible to demonstrate additional enzymatic cascade reactions within these coacervates, directly correlating with the influence of 14-3-3 protein on cellular signal transduction, beyond the current demonstration of peptide exchange and binding with a simple fluorescent readout by the authors?
2. What are some other examples of proteins that could be utilized similarly to 14-3-3 protein as models to highlight the significance of affinity-based protein-protein interactions in chemical communication within the coacervates model, as demonstrated by the authors?

3. What are the potential applications of using the 14-3-3 protein in model coacervates for drug development within the field?

Competitive protein location in artificial cells - response to reviewer's comments

Original comments in black

Author's responses in green

Modifications to manuscript in red

Reviewer #1 (Remarks to the Author):

In this interesting manuscript the authors characterize competitive binding and communication between coacervates utilizing 14-3-3 binding proteins and ligands of differing affinity. Their complex coacervates, also termed 'artificial cells' by the authors, are formed by association of positively charged quaternized amylose and negatively charged carboxymethylated amylose. They incorporate Ni-NTA to tether his-tagged proteins and anchor 14-3-3 protein isoforms, which have distinct affinities for substrates, for example ranging from 25-1000 nM for the c-Raf pS peptide. They demonstrate selective recruitment of target peptides dependent on presence of 14-3-3 domain in the coacervates and demonstrate competitive recruitment in which the coacervates containing the higher affinity binding isoform of 14-3-3 better recruit a FITC-labelled c-Raf pS peptide. Intriguingly, target phosphorylation can selectively effect its binding, causing release from the coacervates. And they set up a shuttling system to either retain or transfer peptide from sender to receiver coacervates. Strengths of the work include demonstration of the validity of conceptual and experimental framework to create peptide competition and communication between compartments in protocell like systems, and demonstration of fast ATP dependent release of target peptides from coacervates. Enthusiasm is somewhat diminished by the extent of competition and from missing controls.

We acknowledge reviewer 1 for their time to evaluate the manuscript and for their comments.

Major Comment

1. Limited extent of competition and missing controls.

In Fig 2F, despite a 40-fold difference in peptide affinity, the authors achieve only ~ 1.5-2-fold enrichment of target in the coacervate containing the strong binding domain. Can they not optimize the reaction conditions and protein concentration to achieve a higher differential partitioning?

We realize that the fold enrichment in the coacervates does not meet expectations based on the fold difference in affinity. We have calculated the distribution of 14-3-3-peptide complexes using a thermodynamic equilibrium model (Figure S4a and S4b). At the estimated local concentrations of 14-3-3 (17.5 μM dimer), we calculated a 34-fold difference in peptide binding between 14-3-3 γ (strong binder) and 14-3-3 σ (weak binder).

Figure S4. a) Local concentration of species at increasing concentration of c-Raf phosphopeptide (c-Raf pS) in a system with 17.5 μM of 14-3-3 σ and 14-3-3 γ dimers (1:1 binding), as calculated by a thermodynamic equilibrium model. The concentration is based on the quantification of 14-3-3 monomer concentration in the coacervates using a fluorescence calibration curve. b) Transformation of the data into molar fractions of complexes of c-Raf pS, and the resulting difference in c-Raf pS uptake between the coacervates with both isoforms. c) Molar fractions of complexes of c-Raf pS, taking into account the 14-3-3 γ that has been transferred to the 14-3-3 σ coacervates (dashed line). Also shown is the calculation of the enrichment of the c-Raf pS peptide into the 14-3-3 γ coacervates. The occupancy was

determined via interpolation of model-generated data using a four-parameter logistic model (GraphPad 10.2.1).

This is indeed lower than what is experimentally found. One possible explanation is that 14-3-3, like the SUMO-c-Raf protein, exchanges between coacervate populations over time. We have re-analyzed the experiment from Figure 2 to explore this effect, and found that 14-3-3 σ is effectively retained in its population with an 18-fold difference in 14-3-3 σ concentration between its initial population and the 14-3-3 γ population. However, 14-3-3 γ , the strongly binding isoform, is more dynamic. It is retained weakly in its initial population, with a final ratio of 5.5-fold between its initial population and the 14-3-3 σ population, as shown in Figure S5 below.

Figure S5. Uncropped images and quantification of 14-3-3 signal related to Figure 2. a) Confocal micrographs of a 1:1 (v/v) mixed sample of coacervates containing 14-3-3 γ or 14-3-3 σ (100 nM) after overnight equilibration. The brightness of the bottom row of images was adjusted for visibility, and

cyan and magenta outlines were added as a visual guide. Scale bar: 25 μm . b) Quantifications of the 14-3-3 isoforms in both coacervate populations from the unedited image.

We hypothesize that the (homo)dimerization affinity plays a role in the different dynamics of the two isoforms. It is known that 14-3-3 σ is the only isoform to exclusively form homodimers in cells and in solution (<https://doi.org/10.4161/cc.5.24.3571>). At the dimerization interface, 14-3-3 σ forms 3 salt bridges whereas 14-3-3 γ only forms 2 salt bridges. It is possible that therefore, the 14-3-3 γ is a weaker dimer in the highly charged coacervate environment and that monomeric 14-3-3 γ is present to a higher degree. We expect a dimeric protein, effectively retained by two His-tags, to be captured in the coacervates in a more stable manner than a monomeric protein. We speculate that 14-3-3 σ , a stronger dimer, is more effectively retained in its coacervate population due to this effect. The weaker dimer 14-3-3 γ , on the other hand, is partly translocated to the 14-3-3 σ -containing population based on more dynamic interactions with the Ni-NTA-amylose. Taking the dynamics into account, our model now predicts a 4.6 fold change in affinity, which indeed is much closer to the observed value (Figure S4c). Hence, in our opinion, the redistribution of 14-3-3 γ over time is the main cause of a diminished change in binding selectivity.

Figures S4 and S5 were added to the Supporting Information. We have also added the fold change to Figure 2f to present readers more quantitative data. The following sentences have been added to the main text:

This 1.9-fold enrichment, however, is lower than the expected fold change (34-fold),³⁷ calculated using a thermodynamic model based on biochemical solution data (Figure S4a and S4b). This can be explained by the partial exchange of 14-3-3 proteins between the coacervate populations (Figures S4c and S5). This exchange is more prominent for 14-3-3 γ than for 14-3-3 σ , because the 14-3-3 σ dimer is a more stable complex than 14-3-3 γ . 14-3-3 σ has 3 salt bridges at its homodimer interface whereas the 14-3-3 γ homodimer only has 2 salt bridges.³⁸ As incorporation of 14-3-3 in the coacervates is governed by interactions of their His-tags with Ni-NTA-Am, the more stable 14-3-3 σ dimer is effectively anchored into the coacervates via a double His tag, whereas the more dynamic 14-3-3 γ will also be present in its monomeric form carrying only one His tag. Thus, a more dynamic 14-3-3 dimerization process could lead to more dynamic protein exchange between coacervates. This indicates the differentiated recruitment of the c-Raf peptide is governed by the degree of exchange of 14-3-3 between coacervates.

Missing controls. ATP is known to act as a hydrotrope and thus for Fig 3D, the proper control is not just no ATP, but rather removal or the phosphorylation site from the c-Raf peptide. And additionally consider changing it to Glu or Asp as a phosphomimic

ATP could indeed function as a hydrotrope, and owing to its negative charge it will likely be recruited in the coacervates effectively. However, we have previously observed a kinase-dependent effect in a different assay format in our previous work (<https://doi.org/10.1002/adma.202300947> - Figure 5). In that experiment, the ATP concentration was kept constant and only the presence of the kinase was varied.

Similarly, we included a control experiment here, where we did not include the kinase but did include ATP (Figure S10). This produced no significant GFP-BiExoS displacement from the coacervates. Figure S10 was added to the Supporting Information along with an accompanying sentence in the main text:

We confirmed that this effect was mediated by kinase activity by performing a control experiment in the absence of kinase, which showed no ATP-dependent effect (Figure S10).

Figure S10. a, b) Control samples for the displacement of GFP-BiExoS by His₁₀-SUMO-c-Raf in the absence of kinase PKA, in the absence (a) or presence (b) of ATP. In both samples, no release of GFP-BiExoS over 30 min was observed. Conditions: 100 nM of 14-3-3γ (Cy5-labeled), 50 nM of His₁₀-SUMO-c-Raf, 50 nM of GFP-BiExoS, bulk concentrations. Scale bar: 25 μm. c, d) Quantification of the GFP-BiExoS signal in micrographs in panels a and b. Statistical differences were analyzed by unpaired t-test, with $N \geq 49$ coacervates across 2 imaging positions in the same sample. Dashed lines represent the median and dotted lines represent the upper and lower quartiles. ns: no statistical difference, **: $p < 0.01$.

Reviewer #2 (Remarks to the Author):

Jan C. M. van Hest and colleagues investigate the control of protein recruitment in artificial cells, with an emphasis on protein exchange across different populations of coacervate-based cells. The work is beautifully written, and the science is sound.

Recommendation: Accept with minor revisions highlighted.

Although the study has some interesting refinements, such as affinity-regulated exchange, competitive binding, intracellular affinity modulation, directed protein transfer, and application of affinity differences, there is very little novelty given that the authors published an article with the exact premise but using different client peptides (Enzymatic Regulation of Protein-Protein Interactions in Artificial Cells, please refer to <https://doi.org/10.1002/adma.202300947>). The novelty of the work has therefore been diluted.

Reviewer 2 is thanked for their time taken to evaluate the manuscript and for their comments.

Other minor concerns that need to be addressed by the authors

In Figure S1, the estimation of protein concentration using fluorescence may be inaccurate. The authors can compare fluorescence-based estimation with other spectrometric methods, like the BSA assay.

This specific experiment mainly serves to provide the reader with an estimation of the protein concentration. Notwithstanding, the obtained value agrees well with the value we previously found (64 μ M local concentration, <https://doi.org/10.1002/adma.202300947>). We acknowledge the reviewer's concerns, that the free dye was used together with a calculation based on the degree of protein labelling. While a BCA assay (we assume the BSA assay mentioned by the reviewer is a typo) is indeed a widely established method for determination of protein concentration, the lower limit of detection is 5 μ g/mL (according to the Thermo Fisher protocol). Given our solution concentration usage of 100 nM of 14-3-3 (30 kDa), which corresponds to 3 μ g/mL, calculation of the internal protein concentration by back-calculation using the dilute phase concentration is not possible. Alternatively, the dense phase would have to be isolated on large volumes, which would correspond to >2.5 mL of coacervate sample. Such an experiment would be extremely costly, and other parts of the coacervate system would also give a signal in the BCA assay, such as the membrane-forming terpolymer and the copper-chelating NTA-amylose. Therefore, we feel that estimation of the protein concentration using a fluorescence calibration curve, of course using the same laser and microscope settings, is most appropriate.

In Fig. 3d, please specify the fluorescence being quantified on the Y axis. In this case, GFP.

We regret this oversight and have updated Figure 3d to specify the species being quantified.

Reviewer #3 (Remarks to the Author):

I co-reviewed this manuscript with one of the reviewers who provided the listed reports. This is part of the Communications Chemistry initiative to facilitate training in peer review and to provide appropriate recognition for Early Career Researchers who co-review manuscripts.

Reviewer 3 is thanked for taking the time to review this manuscript and for providing comments.

Reviewer #4 (Remarks to the Author):

The manuscript by Veldhuisen et al. elucidates the competitive recruitment of a client peptide into the 14-3-3 protein hub within artificial cells. They employ an affinity-regulated exchange method to demonstrate protein-protein interactions (PPIs) involving the 14-3-3 hub protein within diverse populations of coacervate-based artificial cells. The significance of 14-3-3 proteins in living cell signal transduction arises from their multifaceted roles as molecular scaffolds, regulators of enzyme activity, and modulators of protein localization. This versatility enables precise control of cellular signaling pathways and responses to environmental cues. Leveraging the adaptable binding landscape of 14-3-3 scaffold proteins with their binding

partners presents an intriguing opportunity to implement affinity-regulated protein exchange and investigate PPIs in coacervate-based artificial cells. By incorporating different isoforms of 14-3-3 with varying PPI affinities, the system can direct a client peptide to artificial cells with higher binding affinities. Phosphorylation is utilized to manipulate the affinity of client proteins, displacing weaker binding partners from 14-3-3 and releasing them from coacervates. Through affinity engineering and competitive binding, a communication system is established between coacervates, facilitating the transport of client proteins from strongly recruiting artificial cells to those with weaker recruitment capabilities. Demonstrating efficient protein translocation between coacervates suggests the potential for transporting active proteins for synthetic signaling pathways, such as kinases. This system holds promise for advancing our understanding of biological systems, where variations in affinity and competition among binders could influence protein uptake in distinct cells or intracellular compartments.

The utilization of 14-3-3 protein and affinity-based interactions plays a critical role in the dynamics of signal transduction within living cell membranes. This intricate mechanism of protein recruitment not only facilitates targeted uptake but also enables the exchange of client proteins, essential for cellular communication. Replicating this process in artificial cells holds immense promise, offering a means to mimic the sophisticated intercellular chemical communication observed in biological systems. By engineering artificial cells capable of selectively taking up and exchanging proteins, novel avenues for studying intercellular communication and developing biomimetic systems emerge. The ability to engineer coacervates with differentiated behaviors and establish communication systems between them could open up new avenues for studying intercellular communication and developing advanced biomimetic systems for various applications. Therefore, these claims are likely to be of interest to researchers and professionals in the relevant scientific communities, especially in bottom-up synthetic biology and biotechnology field. Conceptually, the idea in the manuscript is straightforward and reasonably laid out. Therefore, I recommend this article for publication in communication chemistry journal.

Reviewer 4 is acknowledged for taking time to evaluate this manuscript and for their comments.

However, I have few minor questions for the authors:

1. Is it feasible to demonstrate additional enzymatic cascade reactions within these coacervates, directly correlating with the influence of 14-3-3 protein on cellular signal transduction, beyond the current demonstration of peptide exchange and binding with a simple fluorescent readout by the authors?

The inclusion of several proteins involved in a signaling pathway in coacervates is something we have also considered. However, this would require the recombinant expression of several proteins that function together, which can be challenging and may require the use of more advanced eukaryotic expression systems. Furthermore, this also requires the design of a readout that accompanies the modulation of the pathway. This is currently out of the scope of the research. We have previously demonstrated the use of bioluminescence as an enzymatic readout (<https://doi.org/10.1002/adma.202300947>), but this is challenging to apply to kinetic measurements such as those presented in this manuscript due to substrate depletion over time. Also, plate reader based readouts do not provide information on the origin of the signal (i.e. in which population the signal is generated). Therefore, we opted to use confocal microscopy as a readout. We have highlighted this choice in the third paragraph of the Results section as follows:

Although we have previously used a bioluminescence assay for PPIs in coacervates,²⁶ confocal microscopy is more suited for determining the localization of a client in a multi-population coacervate sample over time since it additionally provides spatial information and is not enzyme substrate-dependent.

2. What are some other examples of proteins that could be utilized similarly to 14-3-3 protein as models to highlight the significance of affinity-based protein-protein interactions in chemical communication within the coacervates model, as demonstrated by the authors?

The 14-3-3 protein was selected because of its wide range of interaction partners and well-studied interactions. In a similar manner, any competitive or regulatable protein-protein interaction could be studied in the coacervate environment. We consider the class of hub proteins the most interesting for now, and the coacervate environment provides a controlled system to study competitive binding of clients to these hub proteins. Other examples include scaffold proteins LAT and SLP-76, interacting with Src homology domains SH2 and SH3, influenced in T cell signaling. Also, the PDZ scaffold PSD-95, functional in synaptic cell-cell signaling would be an interesting target. We have added the following sentence to the final paragraph of the conclusion:

We also envision the substitution of 14-3-3 with other hub proteins such as PSD-95, which is involved in neuronal cell-cell signaling.

3. What are the potential applications of using the 14-3-3 protein in model coacervates for drug development within the field?

This is an interesting idea. The manuscript presented here focuses on competitive protein-protein interactions and how these influence protein recruitment. Many protein-protein interactions including those of 14-3-3 can be modulated by small molecules. One could indeed foresee the modulation of protein-protein interactions that we demonstrate in this work, although this is out of the scope of the current manuscript. We imagine this would require careful selections of binding partners and chemical matter.

Other modifications by the authors

- We noticed a text label of a microscopy image was not aligned properly in the original Figure S8, now Figure S11, and have aligned it correctly. The data and conclusions drawn from the image remain unchanged.

REVIEWERS' COMMENTS:

Reviewer #1 (Remarks to the Author):

The revised manuscript has been sufficiently improved and addressed my concerns

Reviewer #2 (Remarks to the Author):

Editorial note: This reviewer provided no further comments to the authors.

Reviewer #3 (Remarks to the Author):

I am very satisfied with the response from the authors. I applaud the authors for a job well done and have no further comments.

Reviewer #4 (Remarks to the Author):

The authors appear to have responded to all the new concerns/comments raised by reviewers in a satisfactory manner. I would suggest publication.